# Appraisal of clinical practice guidelines and consensus statements on obstetric anaesthesia: a systematic review using the AGREE II instrument

Lu Huang,[1,2] Na Hu,[1,2] Ling Jiang,[1,2] Xinglong Xiong,[3] Jing Shi ,[3] Dongxu Chen [1,2]

¹Department of Anesthesiology, West China Second University Hospital, Sichuan University, Chengdu, Sichuan, China
²Key Laboratory of Birth Defects and Related Diseases of Women and Children (Sichuan University), Ministry of Education, Chengdu, Sichuan, China
³Department of Anesthesiology, The Affiliated Hospital of Guizhou Medical University, Guiyang, Guizhou, China

**Correspondence to**
Dongxu Chen;
scucdx@foxmail.com

## ABSTRACT

**Objectives** Despite the publication of hundreds of trials on obstetric anaesthesia, the management of these conditions remains suboptimal. We aimed to assess the quality and consistency of guidance documents for obstetric anaesthesia.

**Design** This is a systematic review and quality assessment using the Appraisal of Guidelines for Research and Evaluation (AGREE) II methodology.

**Data sources** Data sources include PubMed and Embase (8 June 2023), three Chinese academic databases, six guideline databases (7 June 2023) and Google and Google scholar (1 August 2023).

**Eligibility criteria** We included the latest version of international and national/regional clinical practice guidelines and consensus statements for the anaesthetic management of pregnant patients during labour, non-operative delivery, operative delivery and selected aspects of perioperative monitoring, postpartum care and analgesia, published in English or Chinese.

**Data extraction and synthesis** Two reviewers independently screened the searched items and extracted data. Four reviewers independently scored documents using AGREE II. Recommendations from all documents were tabulated and visualised in a coloured grid.

**Results** Twenty-two guidance documents (14 clinical practice guidelines and 8 consensus statements) were included. Included documents performed well in the domains of scope and purpose (median 76.4%, IQR 69.4%–79.2%) and clarity of presentation (median 72.2%, IQR 61.1%–80.6%), but were unsatisfactory in applicability (median 21.9%, IQR 13.5%–27.1%) and editorial independence (median 47.9%, IQR 6.3%–73.2%). The majority of obstetric anaesthesia guidelines or consensus centred on different topics. Less than 30% of them specifically addressed the management of obstetric anaesthesia perioperatively. Recommendations were concordant on the perioperative preparation, and on some indications for the choice of anaesthesia method. Substantially different recommendations were provided for some items, especially for preoperative blood type and screen, and for the types and doses of neuraxial administration.

**Conclusions** The methodological quality in guidance documents for obstetric anaesthesia necessitates enhancement. Despite numerous trials in this area,

## STRENGTHS AND LIMITATIONS OF THIS STUDY

⇒ A strength of this review is the comprehensive search strategy to identify clinical practice guidelines for obstetric anaesthesia.

⇒ The review was restricted to documents published in English and Chinese.

⇒ Appraisal of Guidelines for Research and Evaluation (AGREE) II was used to evaluate the quality of guidelines with less attention on detailed recommendations and the potential for unconscious bias arose from the subjective assessment of the documents included.

⇒ Caution is warranted in interpreting the AGREE II results, as the AGREE framework assigns equal weighting to all six domains, irrespective of their individual significance.

evidence gaps persist for specific clinical queries in this field. One potential approach to mitigate these challenges involves the endorsement of standardised guidance development methods and the synthesis of robust clinical evidence, aimed at diminishing difference in recommendations.

## INTRODUCTION

Obstetric anaesthesia refers to labour analgesia, caesarean delivery anaesthesia, pain management during the procedure of labour and delivery, and addressing obstetric emergencies. Epidural analgesia is considered the gold standard of intrapartum analgesia,[1] with approximately 10%–83% of women giving birth under neuraxial anaesthesia.[2] Despite technological advancements, anaesthesiologists still encounter significant challenges in safely administering anaesthesia to pregnant patients. Aligning with the 2021 World Patient Safety Day theme 'Safe Maternal and Newborn Care', the primary goal remains safeguarding the health of both mother and fetus during anaesthesia administration.

To enhance anaesthetic care for obstetric patients, many clinical guidelines and consensus statements have been created by healthcare institutions and professionals. They cover aspects like perianaesthetic assessment, and management for vaginal and caesarean deliveries. For example, the American Society of Anesthesiologists (ASA) released obstetric anaesthesia guidelines in 1998, with updates in 2007 and 2016.[3 4] In China, the consensus on obstetric anaesthesia was initially published in 2007, with subsequent updated in 2017 and 2020.[5 6] Additionally, China published two editions of guidelines for labour analgesia in 2016[7] and 2020.[8]

Despite the variety of documents, current guidelines and consensuses on obstetric anaesthesia and analgesia provide varied recommendations, such as whether require a platelet count, whether to order or require a blood type and screen or cross-match.[4] Furthermore, guideline development may be influenced by the experience and composition of the guideline development groups. Low-quality guidance documents would pose risk to individual patients and communities, and impede the application of guideline recommendations in clinical practice.[9] Some guidelines were not written clearly, which may lead to clinician misunderstanding or not identifying key areas clearly, contributing to misjudged prioritising of clinical actions.[10] Hence, it is essential to summarise and compare existing obstetric anaesthesia guidelines or consensus statements to ensure the safety of both maternal and neonates during obstetric anaesthesia. In this study, we aimed to evaluate the scope, content and consistency of current guidance documents on obstetric anaesthesia and analgesia. The quality of included guidelines and consensus statements were evaluated using the Appraisal of Guidelines for Research and Evaluation (AGREE) II instrument and presented as scores.

## METHODS

This study was developed based on the Preferred Reporting Items for Systematic Reviews and Meta Analyses for Protocols.[11] The protocol of this study was registered prospectively at the International Platform of Registered Systematic Review and Meta-analysis Protocol.

### Literature search and selection criteria

We conducted a systematically search of PubMed and Embase from inception to 8 June 2023, employing a comprehensive search strategy (see online supplemental tables 1 and 2) to identify guidelines and consensus recommendations for anaesthetic management of pregnant during labour, non-operative delivery, operative delivery and selected aspects of perioperative monitoring, postpartum care and analgesia. We also searched three academic databases for Chinese publications (the Chinese Biomedical Literature Database, Wanfang Data and Chinese Medical Association Database; see online supplemental table 3) and six guideline databases from inception to 7 June 2023, using search strategies tailored to different databases (online supplemental table 4). To ensure thoroughness, we conducted additional searched on Google and Google Scholar on 1 August 2023. Additionally, we scanned the reference lists of the most relevant clinical practice guidelines and review articles to identify any additional relevant guidelines. We included the latest versions of all international and national/regional clinical practice guidelines and consensus statements for the obstetric anaesthetic management published in English or Chinese. We defined clinical practice guidelines and consensus statements as documents providing recommendations for patient care, which are derived from a systematic review of existing evidence or from collective opinions of an expert panel.[12] The exclusion criteria were as follows: (1) original investigation, study protocols, comments on existing guidelines or consensus, and conference abstracts or posters; (2) draft documents that are under development or not finalised; (3) previous documents replaced by updated versions from the same organisation.

Two reviewers (LH and DC) independently screened all searched documents. Full texts for potentially relevant guidelines or consensus statements were retrieved and examined for eligibility. Reasons for exclusion were provided for documents excluded during the full-text review (online supplemental table 5). Disagreements were resolved through discussion with a third reviewer (JS).

### Data extraction

The following data were extracted from each included document, including publication year, funding body and evidence base. Additionally, the data encompassed recommendations pertaining to monitoring and management, postpartum care and analgesia. Data were extracted by one investigator and cross-checked by another investigator. In the event of any discrepancies, consensus was reached through discussion and resolution.

### Appraisal of guidance documents

The included documents were evaluated by four independent reviewers (LH, NH, LJ, and DC) using the AGREE II tool.[13] AGREE II is a validated instrument widely used to assess the quality of clinical practice guidelines and consensus statements.[14] The tool consists of 23 items categorised into six domains: scope and purpose, stakeholder involvement, rigour of development, clarity of presentation, applicability and editorial independence. Each item is rated on a seven-point scale, ranging from 1 (Strongly disagree) to 7 (Strongly agree), indicating the level of agreement. The method for each domain appraised is present in supplementary methods. In the current study, all appraisers completed an online training tutorial (available: http://www.center-agree-ii-training-tools/ (Accessed 1 September 2023)) prior to commencing the review process to ensure standardisation. Detailed instruction from the AGREE II User's Manual[13] and objective

scoring criteria (online supplemental table 6) for each item were used to promote scoring consistency.

## Synthesis of recommendations

We meticulously extracted recommendations pertaining to critical clinical queries from a comprehensive set to guidance and consensus documents. Recommendations were extracted by two investigators (LH and NH) and checked by another (XX and DC). To visually represent these recommendations, we used a color-coded grid to illustrate any difference (online supplemental summary of key recommendations).

## Statistical analysis

All data were analysed using the SPSS V.25.0 software. Median and IQR were computed for the domain scores. The inter-rater agreement was computed using the intra-class correlation coefficient (ICC) with a two-way random effects model for the total score was computed. The level of agreement (ICC) was categorised according to widely recognised thresholds: poor (<0.40), fair (0.40–0.59), good (0.60–0.74) or excellent (0.75–1.00).[15] Following the approach described in a previously published article,

a domain is considered to have superior quality if its score exceeds 50%.

## Patient and public involvement

No patient involved.

## RESULTS

### Search results

We retrieved 6921 citations, with 6797 being excluded after screening titles and abstracts for not meeting eligibility criteria (figure 1). We thoroughly assessed the remaining 124 citations and excluded 102 as they did not align with our defined criteria. Ultimately, we included 14 guidelines[4 16–28] and 8 consensus statements.[6–8 29–33]

### Characteristics of included guidelines and consensus statements

A summary of the characteristics of the eligible guidelines and consensus statements is presented in table 1. Fourteen of guidelines[4 16–28] and eight consensus statements[6–8 29–33] with the latest versions were identified, published from 2014 to 2023, involving 15 national or multi-international

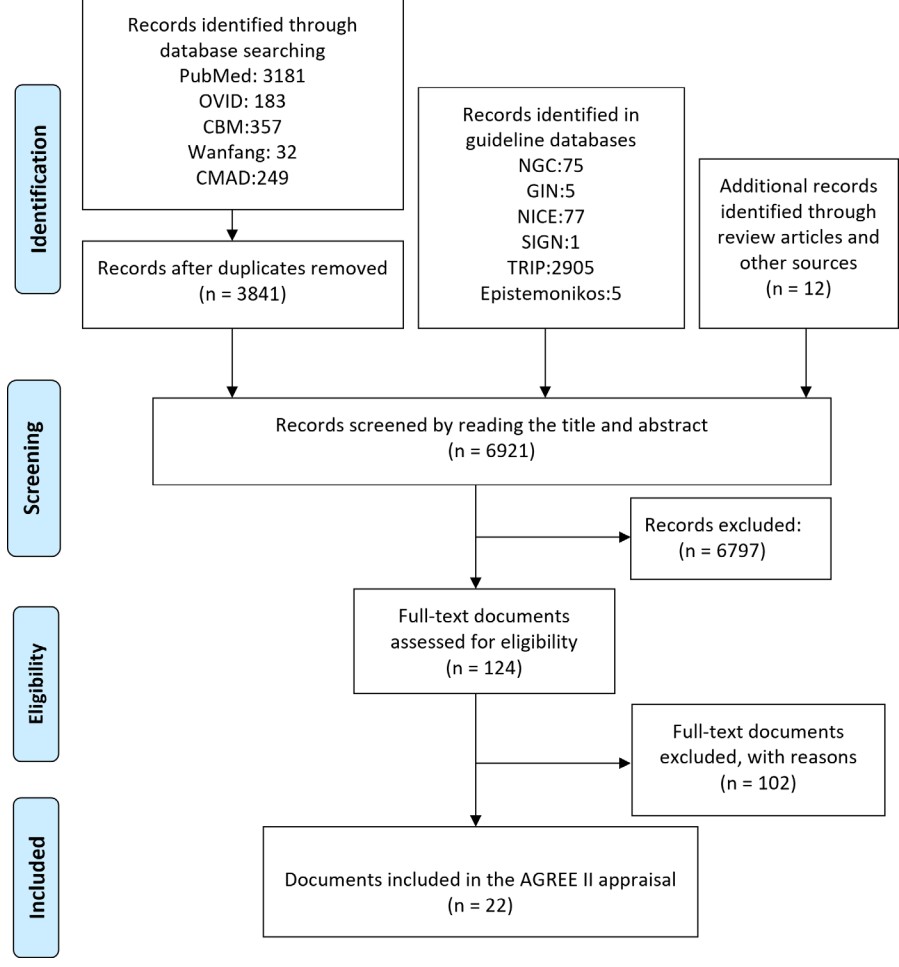

**Figure 1** Flow diagram for literature search. AGREE, Appraisal of Guidelines for Research and Evaluation; CBM, Chinese Biomedical Literature Database; CMAD, Chinese Medical Association Database; GIN, Guidelines International Network; NGC, National Guideline Clearinghouse; NICE, National Institute for Health and Care Excellence; SIGN, Scottish Intercollegiate Guidelines Network; TRIP, Turning Research into Practice Database.

**Table 1** Characteristics of included guidelines and consensus statements

| Document | Issuing organisation | Country | Funding body | Target population | Target audience | Guideline development group | Guideline review | Guideline update | Evidence base | LOE | SOR |
|---|---|---|---|---|---|---|---|---|---|---|---|
| Guidelines | | | | | | | | | | | |
| ASA_2016[4] | ASA | USA | Institutional | Intrapartum and postpartum patients with uncomplicated pregnancies or with common obstetric problems. | Anaesthesiologists | Multi | NG | 1999–2006–2016 | CS and SLR | + | + |
| ERAS_2018 (Part 1, 2, 3)[17 26 27] | ERAS | Multination | NG | Caesarean delivery patients | NG | ERAS | External review | NG | SLR | + | + |
| ACOG_2019[16] | ACOG | USA | Institutional | NG | NG | ACOG | NG | 2007–2013–2017–2019 | NG | + | + |
| SOGC_2019[18] | SOGC | Canada | NG | Women with obesity who are pregnant or planning pregnancies | Healthcare providers | SOGC | External review | 5 years | SLR | + | + |
| SOGC_2022[22] | SOGC | Canada | Institutional | Pregnant women with a dichorionic twin pregnancy | Healthcare providers | SOGC | External review | NG | SLR | + | + |
| AA and OAA_2020[20] | AA and OAA | UK | None | Obstetric neuraxial block patients | NG | AA and OAA | NG | NG | NG | – | – |
| PROSPECT_2020[19] | PROSPECT | Europe | OAA | Caesarean delivery patients under neuraxial anaesthesia | NG | Multi | NG | 2014–2020 | SLR | – | – |
| NICE_2021[21] | NICE | UK | NG | Caesarean delivery patients | NG | NICE | NG | 2011–2014–2021 | SLR | + | + |
| OAA_2022[23] | OAA | UK | None | Caesarean delivery patients | Anaesthetists | OAA | NG | NG | CS and SLR | – | – |
| ANNA_2023[25] | ANNA | USA | NG | Obstetric patients during labour and delivery and related procedures | CRNAs | AANA | NG | 2008–2023 | NG | – | – |

Continued

**Table 1** Continued

| Document | Issuing organisation | Country | Funding body | Target population | Target audience | Guideline development group | Guideline review | Guideline update | Evidence base | LOE | SOR |
|---|---|---|---|---|---|---|---|---|---|---|---|
| RCoA_2023[24] | RCoA | UK | NG | All pregnant women | Healthcare providers | RCoA | External review | Planned in 2024 | NG | + | + |
| French_2021[28] | CARO | France | NG | Caesarean delivery patients | Stakeholders involved in the care of women undergoing caesarean section | Multi | External review | NG | NG | – | – |
| Consensus statements | | | | | | | | | | | |
| OG-OG-CMA_2014[29] | OG-OG-CMA | China | The Research Special Fund for Public Welfare Industry of Health of the Ministry of Health of China | Caesarean delivery patients | NG | OG-OG-CMA | NG | NG | NG | – | – |
| OG-A-CMA_2016[7] | OG-A-CMA | China | NG | Labour analgesia | NG | OG-A-CMA | NG | NG | NG | – | – |
| AAGBI_2017[30] | AAGBI | Europe (Ireland) | None | Caesarean delivery patients | NG | AAGBI | External review | NG | NG | – | – |
| SOAP_2019[33] | SOAP | USA | None | Caesarean delivery patients | NG | SOAP | NG | NG | SLR | + | + |
| CHBSA_2020[31] | CHBSA | China | NG | All pregnant women | NG | CHBSA | NG | NG | NG | – | – |
| CAA_2020[6] | CAA | China | NG | All pregnant women | Anaesthesiologists | CAA | NG | 2008–2017–2020 | NG | – | – |
| CAA_2020[8] | CAA | China | NG | Labour and vaginal delivery | Anaesthesiologists | CAA | NG | NG | NG | – | – |
| SOAP_2020[32] | SOAP | USA | NIH | Caesarean delivery patients | NG | SOAP | NG | NG | NG | + | + |

AA and OAA, Association of Anesthetists and the Obstetric Anaesthetists' Association; AAGBI, Association of Anaesthetists of Great Britain and Ireland; ACOG, American College of Obstetricians and Gynecologists; ANNA, American Association of Nurse Anesthesiology; ASA, American Society of Anesthesiologists; CAA, Chinese Association of Anesthesiology; CARO, Club anesthesie-reanimation en obstetrique; CHBSA, China Healthy Birth Science Association; ERAS, Enhanced Recovery After Surgery; LOE, level of evidence; NICE, National Institute for Health and Care Excellence; OAA, Obstetric Anesthetists' Association; OG-A-CMA, Obstetrics Group, Anesthesiology Branch of Chinese Medical Association; OG-OG-CMA, Obstetrics Group, Obstetrics and Gynecology Branch of Chinese Medical Association; PROSPECT, Procedure-specific postoperative pain management working group; RcoA, Royal College of Anesthetists; SOAP, Society for Obstetric Anesthesia and Perinatology; SOGC, Society of Obstetricians and Gynecologists of Canada.

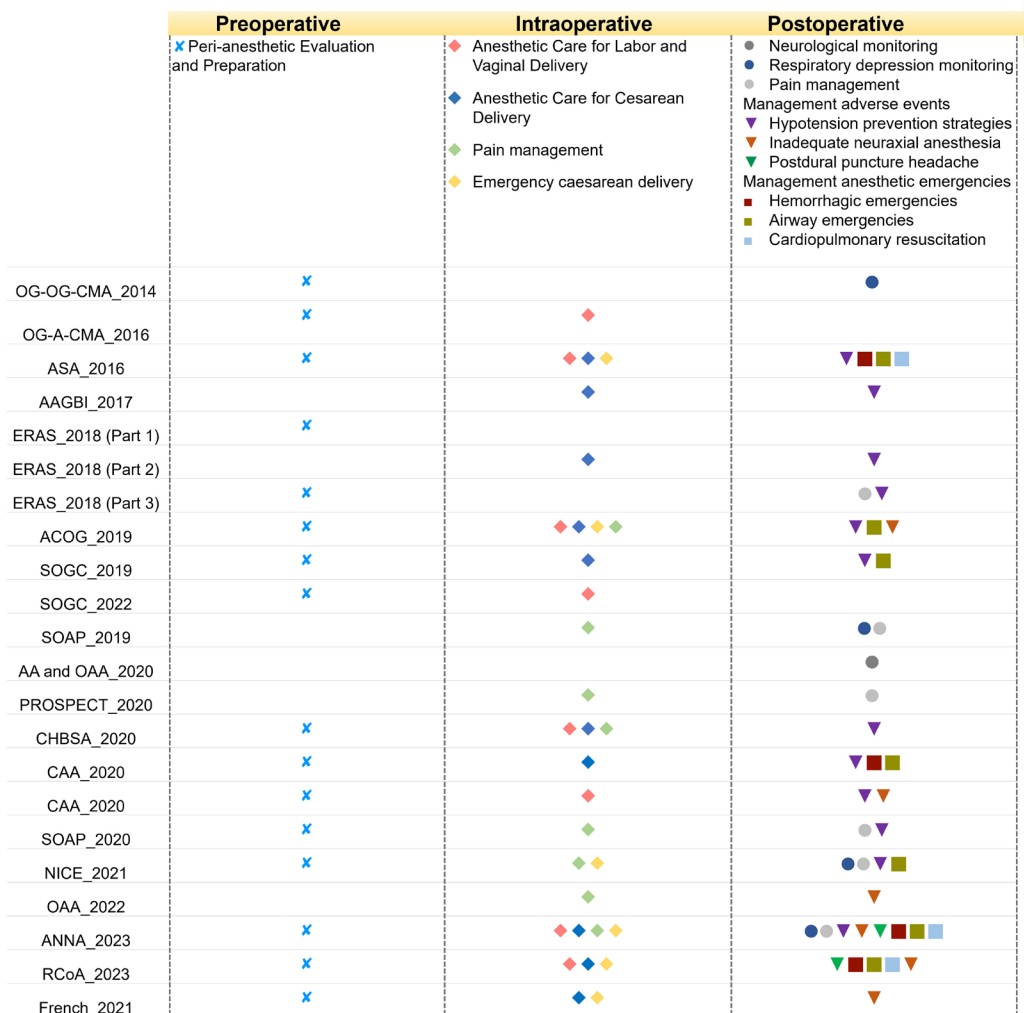

**Figure 2** A summary of the main focus of the included document. AA and OAA, Association of Anesthetists and the Obstetric Anesthetists' Association; ANNA, American Association of Nurse Anesthesiology; AAGBI, Association of Anesthetists of Great Britain and Ireland; ACOG, American College of Obstetricians and Gynecologists; ASA, American Society of Anesthesiologists; CAA, Chinese Association of Anesthesiology; CHBSA, China Healthy Birth Science Association; ERAS, Enhanced Recovery After Surgery; NICE, National Institute for Health and Care Excellence; OAA, Obstetric Anesthetists' Association; OG-A-CMA, Obstetrics Group, Anesthesiology Branch of Chinese Medical Association; OG-OG-CMA, Obstetrics Group, Obstetrics and Gynecology Branch of Chinese Medical Association; PROSPECT, Procedure-specific postoperative pain management working group; RcoA, Royal College of Anesthetists; SOAP, Society for Obstetric Anesthesia and Perinatology; SOGC, Society of Obstetricians and Gynecologists of Canada.

organisations predominantly comprising anaesthesiologists or obstetricians and gynaecologists from the USA, UK, Europe and China. Eight documents[17 18 22 24 26–28 30] clearly stated being externally reviewed. Eight of these guidelines[4 6 16 18 19 21 24 25] included procedures for updating in response to emerging evidence, while only six[4 16 19 22 29 32] disclosed their funding body. The focus of these guidelines was primarily on the obstetric patients undergoing vaginal delivery or caesarean delivery (figure 2), with eight documents[4 6 8 18 22–25] intended for use by anaesthesiologists or other healthcare providers. Additionally, 10 documents[4 17–19 21–23 26 27 33] conducted systematic literature reviews as part of their development process, which 11 documents[4 16–18 21 22 24 26 27 32 33] reported the level of evidence supporting recommendations and grading the strength of the guidance recommendations themselves.

## Appraisal of guidelines and consensus statements

The standardised AGREE II domain scores for each guidance document are presented in figure 3 and provided in value as online supplemental tables 6 and 7. Mean scores for each AGREE II item can be found in online supplemental table 8. The ICC were above 0.90 (online supplemental table 9). The overall quality of guidance documents varied both between documents across domains and within documents across domains. The ASA guideline published in 2016[4] had the highest domain scores, with five domains scoring in the upper quartile. This was followed by the guidelines published by the Enhanced Recovery After Surgery Society Recommendations (part2),[27] American College of Obstetricians and Gynecologists[16] and Royal College of Anesthetists,[24] all with four domains scoring above the upper quartile.

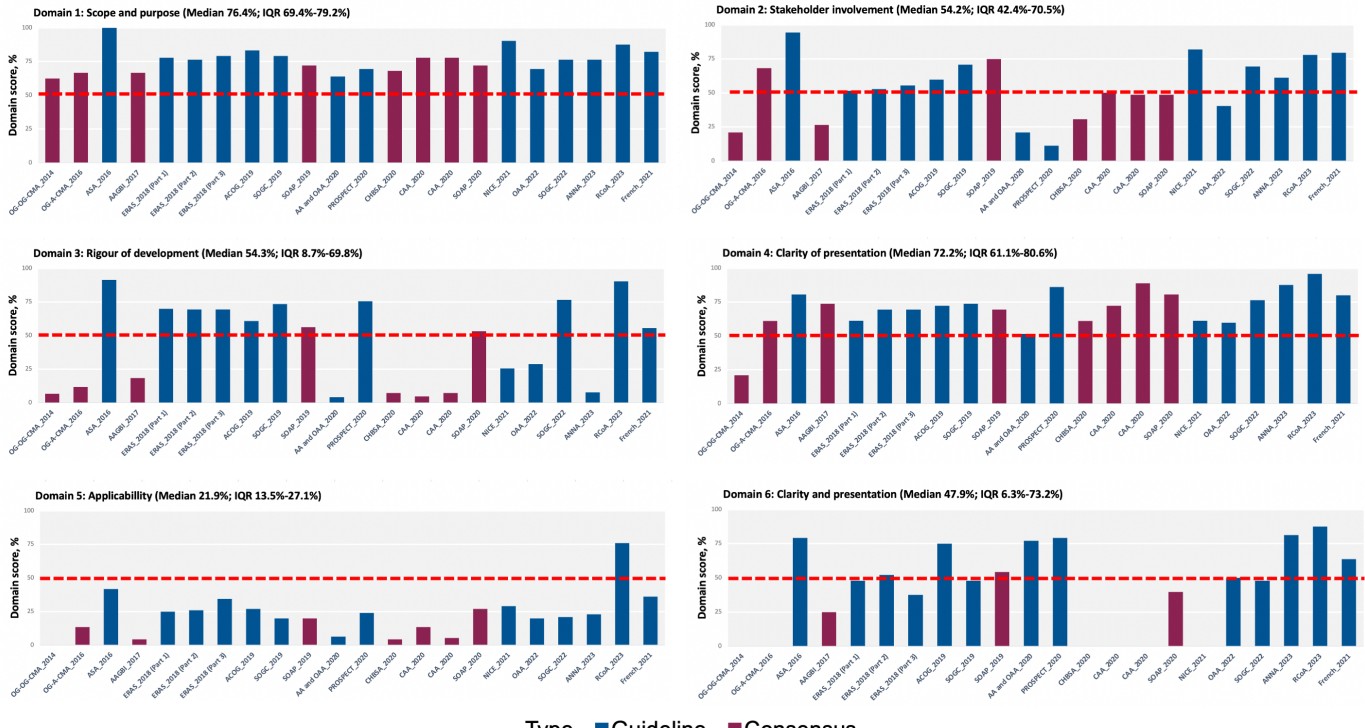

**Figure 3** Standardised domain scores for each document. AA and OAA, Association of Anesthetists and the Obstetric Anesthetists' Association; AAGBI, Association of Anesthetists of Great Britain and Ireland; ACOG, American College of Obstetricians and Gynecologists; ANNA, American Association of Nurse Anesthesiology; ASA, American Society of Anesthesiologists; CAA, Chinese Association of Anesthesiology; CHBSA, China Healthy Birth Science Association; ERAS, Enhanced Recovery After Surgery; NICE, National Institute for Health and Care Excellence; OAA, Obstetric Anesthetists' Association; OG-A-CMA, Obstetrics Group, Anesthesiology Branch of Chinese Medical Association; OG-OG-CMA, Obstetrics Group, Obstetrics and Gynecology Branch of Chinese Medical Association; PROSPECT, Procedure-specific postoperative pain management working group; RcoA, Royal College of Anesthetists; SOAP, Society for Obstetric Anesthesia and Perinatology; SOGC, Society of Obstetricians and Gynecologists of Canada.

In total, 81.8% (18 of 22) of the included documents were of excellent quality. Guidance documents received the highest scores for the scope and purpose domain (median 76.4%, IQR 69.4%–79.2%) and the clarity of presentation domain (median 72.2%, IQR 61.1%–80.6%). However, they received the lowest scores for the applicability domain (median 21.9%, IQR 13.5%–27.1%) and the editorial independence domain (median 47.9%, IQR 6.3%–73.2%). No tendency of improvement in the quality score over time was observed.

### Synthesis of recommendations

#### Approaches to the perianaesthetic evaluation and preparation

Sixteen documents[4 6–8 16–18 21 22 24–26 28 29 31 32] were included in the perianaesthetic evaluation and preparation. Key recommendations are presented in online supplemental table 10. Ten documents[4 6–8 16 24 25 28 29 31] provided recommendations for history and physical examination in pregnant individuals, with an emphasis on maternal health, anaesthetic history and relevant obstetric history. Of these, six documents[4 6–8 16 25] especially recommended examining the patient's back when a neuraxial anaesthetic was planned. Eight documents[4 6 8 16 25 26 28 29] highlighted the important of communication between obstetric providers, and anaesthesiologists to recognise significant anaesthetic or obstetric risk factors.

Six guidelines[4 6 7 16 21 29] referenced in the literature emphasise the important of platelet count in predicting anaesthesia-related complications. Three of these guidelines[4 16 21] recommended individualising the platelet count based on a patient's medical history. Furthermore, five documents[4 6 7 21 29] discussed the necessity of blood type and screen or cross-match, with two of them[4 21] suggesting that routine test is not essential for healthy and uncomplicated parturient (table 2). Additionally, nine documents[4 6–8 16 18 22 25 29] indicated that fetal heart rate patterns should be monitored during the anaesthetic process.

Eleven documents[4 6 17 18 21 24–26 29 31 32] addressed aspiration prevention during delivery. Regarding fasting, five documents[4 6 26 31 32] suggested that clear liquids up to 2 hours before anaesthesia induction, and six documents[4 6 26 29 31 32] recommended a fasting period of 6–8 hours for solids. The use of antacids, $H_2$-receptor antagonists and metoclopramide were also recommended for aspiration prophylaxis during caesarean delivery.[4 6 17 21 24 26 31]

**Table 2** Some different recommendations summarised

| | | The most frequently recommended content | A partially consistent recommendation | Contrary recommendation |
|---|---|---|---|---|
| Approaches to the perianaesthetic evaluation and preparation | | | | |
| Routine intrapartum platelet count | | Agree | – | Individualised |
| Routine blood type and screen | | Agree | – | Individualised |
| Routine blood cross-match | | Agree | – | Individualised |
| Approaches to the anaesthetic care for labour and vaginal delivery | | | | |
| Bupivacaine | Continuous infusion epidural analgesia | 0.04%–0.125% | 0.0625%–0.125% | – |
| | Single-injection spinal | No guideline | 2.0–2.5 mg | – |
| Ropivacaine | Continuous infusion epidural analgesia | 0.0625%–0.15% | 0.08%–0.2% | – |
| | Single-injection spinal | No guideline | 2.3–3 mg | – |
| Fentanyl | Continuous infusion epidural analgesia | 0.5–2 µg/mL | 1–2 µg/mL | – |
| Sufentanyl | Continuous infusion epidural analgesia | 0.2–0.6 µg/mL | 0.4–0.6 µg/mL | – |
| Approaches to the anaesthetic care for caesarean delivery | | | | |
| Bupivacaine | Continuous infusion epidural analgesia | 0.50% | 0.0625%–0.125% | |
| | Single-injection spinal | No guideline | 5–15 mg | – |
| Ropivacaine | Continuous infusion epidural analgesia | 0.5%–0.75% | 0.08%–0.2% | |
| | Single-injection spinal | No guideline | 10–20 mg | – |
| Approaches to the monitoring of obstetric and anaesthetic or intraoperative or postoperative pain management | | | | |
| High-risk maternal respiratory monitoring | | No guideline | q1h clinical assessment for first 12 hours or half hour for 2 hours | – |
| Neuraxial adjuvant drugs in spinal anaesthesia | | 50–150 µg | 150–500 µg | – |
| Neuraxial adjuvant drugs in epidural or combined spinal-epidural technique. | | 2–3 mg | 2.5–5 mg | – |
| Approaches to the adverse events during obstetric anaesthesia | | | | |
| Vasoactive drugs used for preventing hypotension | | 20–50 µg or 0.5 µg/kg/min | 20–50 µg or 0.25/kg/min | – |
| | | No guideline | 1–2 mg or 4 µg/kg/min or 0.5 mg/mL | – |
| | | 4–6 µg or 0.08 µg/kg/min | – | – |
| | | 2 to 15 mg intravenous | 2–4 mg | |

Note: it is paramount to emphasise that the above contradictory recommendations serve as a concise overview, without claiming comprehensiveness.

Five documents[7 8 16 24 25] discussed the use of heparin or low-molecular-weight heparin during the perianaesthesia and post anaesthesia, and the ACOG[16] explicitly stated that prophylactic subcutaneous unfractionated heparin at a dosage of 5000 units two times a day did not contraindicate neuraxial techniques, as long as catheter placement or removal more than 4–6 hours since last dose. In the case of intermediate-dose unfractionated heparin (7500–10 000 units) more than 12 hours after the last dose or high-dose unfractionated heparin (total daily dose greater than 20 000 units) more than 24 hours after the last dose with activated partial thromboplastin time, it was deemed low risk to proceed with neuraxial anaesthesia.

### Approaches to the anaesthetic care for labour and vaginal delivery

Eight documents[4 7 8 16 22 24 25 31] addressed anaesthetic care for labour and vaginal delivery (summarising in online supplemental table 11). Three documents[4 7 8] explicitly indicated that neuraxial analgesia could be administrated based on the level of discomfort the parturient, without limitation of cervical dilation. The ASA suggested providing labour neuraxial analgesia for parturient in early labour (ie, cervical dilations of less than 5 cm dilation).[4] These documents[4 7 8] also recommended to offer neuraxial analgesia on an individualised basis. The ASA also provided recommendations on the administration of neuraxial techniques for patients attempting vaginal birth after previous caesarean delivery, as well as early placement of a neuraxial catheter.[4]

There were three methods for neuraxial anaesthetic: continuous infusion epidural analgesia (CIEA), combined spinal–epidural analgesia (CSEA) and single-injection spinal. Five documents[4 7 8 16 25] covered both CIEA and

CSEA and recommended using local anaesthetic, as well as adding opioids. Before a neuraxial procedure, the platelet count was required more than $70*10^9$/L to reduce the risk of haematoma.[8 16 25] The puncture location was recommended on L2-3 or L3-4 in China published consensus.[7 8] Local anaesthetic including bupivacaine, ropivacaine and levobupivacaine were used with loading dose or maintenance doses. In China, published consensus[7 8] recommended using 0.04%–0.125% bupivacaine or 0.0625%–0.15% ropivacaine to initiate analgesia and maintain with 0.05%–0.125% bupivacaine or 0.0625%–0.15% ropivacaine (table 2). The Chinese Associations of Anesthesiology (CAA) consensus also provided recommendations for 0.04%–0.125% levobupivacaine for loading dose and 0.05%–0.125% levobupivacaine for maintenance.[8] These two documents suggested that opioids, including fentanyl or sufentanyl, could be injected with local anaesthetic, with recommended loading dose of fentanyl was 0.5–2 µg/mL and 1–2 µg/mL for maintenance, and the loading dose of sufentanyl was 0.2–0.6 µg/mL with maintenance dose 0.3–0.6 µg/mL.[7 8] The American Association of Nurse Anesthesiology (ANNA) guideline suggested using 0.0625%–0.125% bupivacaine or 0.08%–0.2% ropivacaine with or without opioids in CIEA.[25] In CSEA, opioids and local anaesthetics could be delivered in spinal, with two recommended protocols[7 8 16 25]: sufentanyl or ropivacaine used singly or in combination. The single dose of sufentanyl was 2.5–7 µg, single dose of ropivacaine was 2.3–3 mg and dose of combined sufentanyl and ropivacaine was 2.5 µg/2.5 mg, respectively.[7 8] Another protocol was that fentanyl and bupivacaine were used singly or with combination. The single dose of fentanyl was 15–25 µg, single dose of bupivacaine was 2.0–2.5 mg and dose of combined fentanyl and bupivacaine was 2.5 µg/2.0 mg, respectively.[7 8] For CSEA and single-injection spinal, both documents recommended using a small-gauge, non-cutting, pencil point needle to access the subarachnoid space.[4 25] Additionally, five documents[4 7 8 25 31] provided recommendations for using patient-controlled epidural analgesia for labour analgesia.

### Approaches to the anaesthetic care for cesarean delivery

Ten documents[4 6 16 18 24 25 27–29 31] covered the anaesthetic care including neuraxial anaesthesia and general anaesthesia for caesarean delivery (summarising in online supplemental table 12). The recommendations for neuraxial anaesthesia including CIEA, single-injection spinal and CSEA were consistent with those for labour and vaginal delivery. Eight of the documents[4 6 16 18 24 25 27 28 31] covered CIEA, CSEA and single spinal injection, with four documents[6 16 25 28] recommended the use of local anaesthetic with or without opioids. A platelet count more than $70*10^9$/L was recommended to reduce the risk of haematoma.[6 16 25] Puncture locations were recommended as follows: L1-2 or L2-3 for CIEA, L3-4 for single injection in spinal and L3-4 in single space for CSEA, while in double space, the puncture location was recommended

in T12-L2 or L3-5.[6 25] To test the effect of neuraxial anaesthesia, the recommended level of block was T6-4.[6] Various local anaesthetic, including lidocaine, bupivacaine, ropivacaine and levobupivacaine, were recommended for use in CIEA. In CAA 2020,[6] lidocaine was recommended with 1.5%–2% concentration, bupivacaine with 0.5% and ropivacaine with 0.5%–0.75% (table 2). And 3% chloroprocaine was also recommended. In contrast, ANNA[25] recommended lower concentrations, with 0.0625%–0.125% for bupivacaine and 0.08%–0.2% for ropivacaine. For single-injection spinal procedures, the recommended doses were 5–15 mg bupivacaine or 10–20 mg ropivacaine.[6 28] To prevent local anaesthetic poisoning, it was recommended that anaesthesiologist withdraw before injection and administer a test dose.[6 16] It was also suggested to add epinephrine to local anaesthetic solutions to delay absorption and increase duration of blockade through vasoconstriction of the blood vessels in the area.[6 16] Furthermore, the use of a small-gauge, non-cutting, pencil point needle was recommended.[4 6 25]

Additionally, eight documents[4 6 16 21 24 25 29 31] provided recommendations on general anaesthesia for caesarean delivery. According to the ASA,[4] general anaesthesia may be the most appropriate choice in cases of profound fetal bradycardia, ruptured uterus, severe haemorrhage, severe placental abruption, umbilical cord prolapse and preterm footling breech. Other documents[16 25] recommended the use of general anaesthesia if neuraxial anaesthesia placement was unsuccessful. Induction delivery was recommended within 10 min, with specific intravenous anaesthetics. CAA[6] detailly recommended intravenous anaesthetics including thiopental sodium (4–5 mg/kg), propofol (1.5–2.5 mg/kg), etomidate (0.2–0.3 mg/kg) and ketamine (0.5–1 mg/kg) as induction agents. Fentanyl (2–5 µg/kg), sufentanyl (0.2–0.5 µg/kg) and remifentanil (4 ng/kg for target) were commonly applied in the general anaesthesia. Succinylcholine (1–1.5 mg/kg) or rocuronium (0.6–1.2 mg/kg) was used as a muscle relaxant.

### Approaches to the monitoring of obstetric and anaesthetic or intraoperative or postoperative pain management

To mitigate postcaesarean delivery pain, nine documents[16 17 19 21 23 25 31–33] provided recommendations for the management of intraoperative or postoperative pain (summarising in online supplemental table 13). Six of these documents[16 19 21 25 31–33] recommended the administration of intrathecal morphine 50–150 µg or epidural 1–3 mg for analgesia. Two documents[19 21] also suggested the use of epidural diamorphine (2–3 mg or 2.5–5 mg) for analgesia. ANNA[25] recommended hydromorphone 75–150 µg in intrathecal administration as well. When morphine was used in neuraxial block, four documents[21 25 29 33] suggested monitoring respiratory frequency and modality. The Society of Obstetric Anesthesia and Perinatology[33] explicitly stated that using morphine less than 0.15 mg or epidural morphine less than 3 mg in healthy parturient would pose low risk,

and recommended monitoring respiratory rate and sedation measurement every 2 hours for 12 hours postoperatively. However, in high-risk circumstances such as cardiopulmonary or neurological comorbidity, obesity (body mass index $\geq$40 kg/m$^2$), obstructive sleep apnoea, chronic opioid use or abuse, and hypertension, or who was administrated with magnesium, as well as high doses of morphine (morphine greater than 0.15 mg or epidural morphine greater than 3 mg), more intensive monitoring (respiratory rate and sedation assessments were monitored for every 1 hour for first 12 hours, every 2 hour for 12–24 hours) and additional modalities such as pulse oximetry and capnography were suggested (table 2). To minimise systemic opioids utilisation, eight documents[16 17 19 21 25 31–33] recommended the administration of non-steroidal anti-inflammatory drug for postoperative pain. Two guidelines[19 25] suggested a single dose of intravenous dexamethasone for analgesia. Local anaesthetic infiltration for the wound after caesarean delivery was recommended by six documents,[16 17 19 25 31 32] and continuous local anaesthetic infusion with the wound was recommended by three documents.[16 19 31] Nerve blocks, including transversus abdominis plane block and erector spinae plane block, for improving postcaesarean delivery pain was recommended by five documents.[16 17 19 31 32] Patient-controlled intravenous analgesia used for pain management after caesarean was recommended by four documents.[16 19 21 31] Regarding obstetric neuraxial block, the Association of Anesthetists and the Obstetric Anesthetists' Association (AA and OAA)[20] stated that straight-leg raising was used for neurological monitoring, and if a maternal was unable to straight-leg raise at 4 hours from the last dose of epidural/spinal local anaesthetic, neurological injury required be investigated.

### Approaches to the adverse events during obstetric anaesthesia

Twelve documents[4 6 8 16–18 21 25 27 30–32] provided recommendations for preventing the hypotension during obstetric anaesthesia (summarising in online supplemental table 14). Eleven documents[4 6 8 16–18 21 25 27 30 31] advised on using 500–1000 mL volume expansion with crystals or colloids, as well as seven documents[4 6 8 21 25 30 31] recommended using left uterine displacement once the maternal was positioned supine for surgery to reduce inferior vena cava compression. Vasopressor management during neuraxial anaesthesia was recommended by 10 documents,[4 6 8 16 17 21 25 30–32] with vasoactive drugs such as phenylephrine, metaraminol, noradrenalin and ephedrine. Phenylephrine with 20–50 µg/min or 0.25–0.5 µg/kg/min was recommended by nine documents.[4 6 8 16 17 25 30–32] Ephedrine 2–15 mg intravenous was recommended by six documents[6 8 16 17 30 31] (table 2). The Association of Anesthetists of Great Britain and Ireland[30] recommended that metaraminol with 0.5 mg/mL could be used to mediate blood pressure. CAA[6] also recommended that noradrenalin 4–6 µg or 0.08 µg/kg/min was useful for preventing hypotension. The target of blood pressure was to maintain systolic arterial pressure greater than 90% and avoid

a decrease to <80% baseline.[21 30] For maternal with preeclampsia, two documents[6 30] advised that vasopressor infusion should not be prophylactically used.

Six documents[8 16 23–25 28] provided recommendations for inadequate neuraxial anaesthesia, and three guidelines[23 25 28] suggested assessing other causes of pain, evaluating the catheter site, and reassessing the sensory block level. For rescuing inadequate anaesthesia, four documents[16 23 25 28] provided recommendations including administrating a second neuraxial technique or extending the neuraxial technique, repeating boluses of fast-acting opioids (fentanyl 25–50 µg, alfentanil 250–500 µg) or ketamine (10 mg boluses), and if necessary, recommending general anaesthesia.

Unintentional dural puncture would lead to postdural puncture headache (PDPH). AANA[25] provided recommendations on using small, non-cutting spinal needle or placing intrathecal catheter for preventing PDPH. Treatment options for PDPH included epidural blood patch, adequate hydration, multimodal analgesia, caffeine, sumatriptan, gabapentin and sphenopalatine ganglion block.

### Approaches to the management of obstetric and anaesthetic emergencies

Four documents[4 6 24 25] provided recommendations for managing haemorrhagic emergencies, with three of them[4 6 25] consistently recommending the use of large bore intravenous catheters (summarising in online supplemental table 15). The ASA[4] and ANNA[25] suggested the implementation of a massive transfusion protocol. Additionally, the ASA[4] recommended the availability of blood bank resource, fluid warmer, forced-air body warmer and equipment for rapid infusion of intravenous fluids and blood products.

Seven documents[4 6 16 18 21 24 25] addressed airway emergency in obstetric anaesthesia, with three of them[4 16 25] recommending the availability of personnel and equipment, such as laryngoscope with assorted blades and endotracheal tubes with stylets. In cases where tracheal intubation fails, the use of emergency procedures such as cricothyrotomy was advised. Access to oxygen source and suction source with tubing were also recommended.[4 21 25] The ASA provided specific recommendations for medications to support blood pressure, muscle relaxation and hypnosis.[4]

Three documents[4 24 25] covered cardiopulmonary resuscitation, with both the ASA[4] and ANNA[25] recommending the availability of basic and advanced life-support equipment during cardiac arrest. It was also recommended to maintain uterine left displacement during cardiac arrest. If circulation was not restored within 4–5 min, two guidelines suggested performing a caesarean section.[4 24 25]

Six documents[4 16 21 24 25 28] addressed emergency caesarean delivery, with the National Institute for Health and Care Excellence[21] providing specific recommendations for the time of unplanned and emergency caesarean births when there is a threat to the life of the woman or

fetus, within 30 min. The use of neuraxial anaesthesia and general anaesthesia was recommended for emergency caesarean delivery.[16 21 28]

## DISCUSSION

In this study, we systematically reviewed 14 guidelines[4 16–28] and 8 consensus statements.[6–8 29–33] The comparison of the content of the 22 documents showed that several documents were very comprehensive, covering many aspects of obstetric anaesthesia,[4 6–8 16 18 21 24 25 27 28 31 32] while other documents may focus on the specific contents.[17 19 20 22 23 26 29 30 33] We found generally average methodological quality and some varied recommendations from included documents covering the preoperative screening and management of obstetric anaesthesia and analgesia. We summarised all key recommendations, and compared and visualised the difference among them, providing a concise but informative overview for clinicians and researchers.

### Comparison with existing research

Our study systematically incorporated both guidelines and consensus statements in the field of obstetric anaesthesia and analgesia, and we observed varied performance across different AGREE II domains for both types of documents. The assessment of the guidelines using the AGREE II tool revealed that the documents assessed in our study excelled in the domains of scope and purpose (domain 1) and clarity of presentation (domain 4), but performed poorly in the domain of applicability (domain 5) and editorial independence (domain 6). This distribution of AGREE II domain scores has been observed in previous guideline appraisal studies, with documents scoring higher in the scope and purpose domain and the clarity of presentation domain, and lower in the applicability domain and the editorial independence domain. This pattern was not only observed in guidance documents for endocrinology diseases,[34 35] but also in documents for other clinical specialties.[36 37] Despite generally low and varied scores in the applicability domain, guidance documents for obstetric anaesthesia and analgesia performed notably worse compared with documents for other conditions,[34 36] indicating that improving the applicability of guidance in this field may be more challenging. One major barrier to achieving good applicability of guidance documents is the time and cost required to perform economic evaluations and pilot studies. A stable and long-term task force for guideline development is needed to conduct these evaluations and studies. While forming such a task force may be difficult in some regions and countries, guidance documents should at least inform the audience about the need to consider these issues. Low scores in the editorial independence domain often resulted from a lack of detailed information on the influence of the funding body and conflicts of interest, with 45.5% of included documents in the current study failing to provide this information. It is essential to disclose the potential influence of the pharmaceutical industry on the synthesis of clinical guidance and promote transparency in financial declarations.

### Gaps, clinical implications and future research

Current guidance and consensus documents uniformly endorse thorough perioperative evaluations and tailored anaesthesia methodologies for delivery, including caesarean sections. Notwithstanding these overarching agreements, variances particularly in areas characterised by high uncertainty persist, likely stemming from the variable quality of the documents, gaps in current evidence or reliance on antiquated studies. This phenomenon is commonplace across all medical disciplines, including obstetric anaesthesia. It underscores the complexity of clinical decision-making and the evolving nature of medical knowledge. Thus, continuous learning and future research endeavours are essential in narrowing the gap and ensuring optimal patient care in obstetric anaesthesia and beyond.

It is important to acknowledge that discrepancies in recommendations could stem from differing practices or organisational approaches internationally. For instance, some countries initiate blood transfusions following a type and screen with different procedures.[38] It is also imperative to recognise that numerous clinical considerations remain underrepresented in these guidance and consensus documents. Among these, the postneuraxial anaesthesia evaluation of nerve blocks warrants particular attention. This includes the necessity for continuous postoperative sensory monitoring, vigilant respiratory depression surveillance and strategies for addressing inadequate analgesia.

In obstetric practice, neuraxial anaesthesia is a cornerstone of pain management. However, it is not without risk. Atypical presentations of motor or sensory block, such as unexpectedly dense or prolonged effects, may herald complications including unintended intrathecal placement or, in rare instances, neurological pathologies. Our review reveals that among the documents evaluated, only the AA and OAA[20] provided recommendations for neurological monitoring, specifically endorsing the straight-leg raise test. There is a conspicuous need for enhanced vigilance in postprocedure monitoring, particularly in the immediate postpartum period. Current clinical practice often overlooks the importance of neuraxial monitoring; a survey conducted in the UK highlighted that only 56% of units had a formal protocol for postoperative neurological assessment, with a few units deferring evaluation until after the first postdelivery day.[39] This discrepancy indicates that despite adequate guideline recommendations, clinical care often fails to meet expected standards. Therefore, when well-supported guidelines are introduced, it is crucial to intensify efforts to enhance the clinical adoption process. Shortening the timeframe for unified practice and enhancing the implementation efficacy of these guidelines can be achieved through the development of clinical practice pathways and the organisation of training sessions to disseminate guideline

updates. Additionally, greater emphasis should be placed on promoting the implementation of these guidelines. As the adage goes, there is much more to be gained in improving global health by applying existing knowledge than by investing in the development of new drugs.

Our review critically examined respiratory depression monitoring in the context of neuraxial anaesthesia, particularly when neuraxial opioids such as morphine, fentanyl, sufentanyl and remifentanil are administered. The ASA guideline[4] recommended continuous monitored of ventilation, oxygenation and consciousness in patients receiving neuraxial opioids. They further suggested intensified monitoring for those with a heightened risk of respiratory depression. Although the widespread endorsement of opioids in neuraxial anaesthesia, it seemed that the monitoring of respiratory depression was not sufficiently highlight by these guidance documents.

Neuraxial anaesthesia, while widely employed, is not always successful. The definition of 'inadequate' varies within the literature, contributing to this broad prevalence range. The prevalence of inadequate neuraxial anaesthesia ranged from 10.2% to 30.3%[40] although there were several guidelines[8 16 23–25 28] that provided some recommendations on the management of inadequate neuraxial anaesthesia. Patel *et al*[41] revealed that only 23% of clinicians have received formal training in management of inadequate neuraxial anaesthesia. Notably, the literature indices a significant gap in both the identification and management of inadequate neuraxial analgesia. Future initiatives must prioritise the development and dissemination of standardised education and training protocols to effectively recognised and manage inadequate neuraxial anaesthesia.

Emergency caesarean section is often performed in cases with life-threatening conditions for the mother and fetus. Decision-to-delivery intervals was recommended within 30 min in category 1 caesarean birth when there was immediate threat to the life of the woman or fetus, and within 75 min in category 2 caesarean birth when there was maternal or fetal compromise which is not immediately life-threatening.[21] Nevertheless, studies found that decision-to-delivery intervals were frequently unmet.[42 43] There is significant heterogeneity in how decision-to-delivery intervals are defined and measured.[9] A pressing concern for anaesthesiologists is the selection of anaesthesia type—neuraxial anaesthesia versus general anaesthesia—in these emergent scenarios, for which the guidelines lack specific, universally applicable recommendations.

## Limitations

First, our inclusion of only English and Chinese documents may have overlooked key guidance from non-English/Chinese speaking regions, though we sought English versions where possible. Second, we focused on the anesthetic management for pregnant, some specialty guidelines such as for cardiac arrest[44] whose targeting population was involved in pregnant were not included.

Third, unconscious bias might have affected our subjective AGREE II ratings, warranting caution in their clinical application. The subjective nature of AGREE II scoring also challenges the reproducibility and comparison of different reviews. Lastly, certain topics in obstetric anaesthesia, like epidural-related maternal fever, postpartum depression and intrapartum hypothermia, were not covered.

## Conclusions

The current obstetric anaesthesia and analgesia guidelines require significant improvements in methodological quality. They showed inconsistencies in recommendations, highlighting the need for standardised guideline development and high-quality evidence synthesis. Crucially, areas like neuraxial analgesia management and monitoring of postoperative neurological functions and intrapartum fever are under-represented, indicating a gap in evidence-based guidelines and necessitating comprehensive research.

**Contributors** DC designed the study. LH and DC performed the literature search and screening and extracted the data and they are aslo the guarantors of the review. All authors conducted the data analyses. LH and DC created the figures and tables and drafted the manuscript. All authors participated in the interpretation of results, and critically revised the manuscript. All authors approved the final version of the manuscript for publication.

**Funding** This study was funded, in part, by a grant from the Science and Technology Department of Sichuan Province (No. 2023NSFSC1570 and 2024NSFSC1679).

**Competing interests** None declared.

**Patient and public involvement** Patients and/or the public were not involved in the design, or conduct, or reporting, or dissemination plans of this research.

**Patient consent for publication** Not applicable.

**Provenance and peer review** Not commissioned; externally peer reviewed.

**Data availability statement** All data relevant to the study are included in the article or uploaded as supplementary information.

**ORCID iDs**
Jing Shi http://orcid.org/0000-0002-9687-8592
Dongxu Chen http://orcid.org/0000-0002-1414-3163

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
