## [Reviewer comments · BMJ Open]

ARTICLE DETAILS

TITLE (PROVISIONAL)	Appraisal of clinical practice guidelines and consensus statements on obstetric anesthesia: A systematic review using the AGREE II instrument
AUTHORS	Huang, Lu; Hu, Na; Jiang, Ling; Xiong, Xinglong; Shi, Jing; Chen, Dongxu

VERSION 1 – REVIEW

REVIEWER	Benhamou, Dan Paris south university hospitals, Anesthesia and Intensive Care Medicine
REVIEW RETURNED	11-Feb-2024

GENERAL COMMENTS	I have read with interest this submission which analyses and compares several guidelines/recommendations published in the field of obstetric anaesthesia. The authors should be congratulated for their thorough review and hard work to provide this analysis. I have however many concerns with the methods used the interpretation of results. It is unclear why this topic has been chosen as evolution of thinking and care has not seen any revolutionary or major change in the last years, i.e. knowledge and practice have progressed “smoothly” and apparently similarly to many other domains. In the first sentence of the abstract, the authors state that “despite the publication of hundreds of trials on obstetric anaesthesia, the management of these conditions remains suboptimal »... In the main text, the first paragraph provides an example of what could be considered inadequate practice and cites the high rate of caesarean section. Unfortunately this is not obstetric anaesthesia but simply obstetrics and anaesthetists have no role in modifying this rate. There is therefore an initial flaw in the article. A second major problem is that even if recommendations are adequate, care often remains substandard and this is not the fault of recommendations. For example, it is often stated that many years (up to 17 years) elapse between a recommendation is published and its implementation in usual care. Another example can be found in the plethora of studies which compare the content of recommendations and their clinical application. A major gap has been consistently shown, with often only around 50 % of patients receiving adequate care. To paraphrase Lucian Leape, there is more to get (i.e. to improve health in the world) by applying what we already know than by spending money in searching new drugs. This gap is related to another and different problem which relates to dissemination and uptake of recommendations, i.e. implementation. This has led to create a new discipline, i.e. « implementation science » , which has also its own Journal.
---

This misinterpretation comes back again in the Discussion section (P24) when the authors show that a recommendation about neurological monitoring has been published in the UK but is not well followed in practice.

Third, it is difficult to understand why the authors limited their search to papers published in English and Chinese. Although China is a vast country, recommendations published in Chinese cannot be applied elsewhere because language is too different. If one wants to include Chinese recommendations, why not include also those in Spanish or in French for example?

Recommendations were searched from the inception of PubMed to the present days. This is very surprising as recommendations published in the eighties are likely very different from those published in 2023, given the flow of new information gathered in the meantime. Differences would not reflect inconsistencies but simply changes in knowledge.

Looking at some domains of obstetric anaesthesia, it was found surprising that the list used for their review is not complete. As a French physician, I looked if recommendations originating from France (published in English in Journals which can be found in PubMed) and surprisingly, I could not find any of them. For example, a Guideline which dealt with inadequate anaesthesia during caesarean section was published one year before (i.e. in the same period of time) the UK guideline (your ref 25) (doi: 10.1016/j.accpm.2021.100934. Epub 2021 Aug 13. Erratum in: Anaesth Crit Care Pain Med. 2021 Oct 17;40(6):100954. PMID: 34400388). It would have been interesting to compare their results and see if inconsistencies can be found.

Other French Guidelines (i.e. for example on preeclampsia) have been recently published in English but were not included. What the authors of the present review call “inconsistencies” are rather related to domains in which scientific work has not yet provided a clear response. In these cases, it is obvious that human factors can lead to somewhat different views. In decision making processes in which uncertainty is high, what is most important is that reasoning follows a clear and logical process and the final decision might differ between teams which have otherwise a high-quality clinical practice.

In some domains, national Guidelines exist but in some others, consistency is obvious as only one is “the” recommendation which is accepted all over the world. This is the case for cardiac arrest during pregnancy where the ERC/ILCOR recommendations are used worldwide. Surprisingly, these recommendations are not mentioned.

One domain which is discussed here has recently appeared, i.e. enhanced recovery after surgery (ERAS). It is believed that all the recommendations in this domain are very consistent, although they have slightly evolved in the last ten years, for example with a more recent inclusion of prehabilitation.

When inconsistencies exist, they may also be related to differences in practice or organisations across countries. For example in some countries, blood transfusion is preceded by type and screen whereas in other (such as France), blood is verified through a different procedure.

It is unclear to me how did the authors decide that a given recommendation is poor in terms of “applicability” and “editorial independence”. Regarding this last domain, I was again surprised as most recommendations were written by scientific societies which independence is high and also because obstetric anaesthesia is poorly funded by industry, reducing the risk of bias.

REVIEWER	Gómez-Río, Manuel Complejo Hospitalario Universitario A Coruna, Anaesthesiology and Perioperative Medicine
REVIEW RETURNED	11-Feb-2024
GENERAL COMMENTS	I congratulate the authors on their work. The manuscript follows a correct methodology and clear presentation. I simply recommend reviewing the possible errors that I detail below: P12L53: "fetal" instead of "fatal". P14L25 and P15L56: "more than" ... instead of "less than" ... P23L28 I suggest adding the term "Gaps" to the heading: "Gaps, clinical implications and future research". Good luck

VERSION 1 – AUTHOR RESPONSE

Response to Reviewers

Reviewer-1

Comment 1: It is unclear why this topic has been chosen as evolution of thinking and care has not seen any revolutionary or major change in the last years, i.e. knowledge and practice have progressed “smoothly” and apparently similarly to many other domains.

Response: Thank you for your insightful comments. Although there have been no substantial revisions to obstetric anesthesia guidelines recently, this study elucidates the variances among existing guidelines. More critically, it offers a quantitative evaluation of these guidelines, furnishing comprehensive recommendations. These contributions are essential for the practical application of these guidelines in clinical settings.

Comment 2: In the first sentence of the abstract, the authors state that “despite the publication of hundreds of trials on obstetric anesthesia, the management of these conditions remains suboptimal »... In the main text, the first paragraph provides an example of what could be considered inadequate practice and cites the high rate of caesarean section. Unfortunately this is not obstetric anesthesia but simply obstetrics and anesthesiologists have no role in modifying this rate. There is therefore an initial flaw in the article.

Response: We thank the reviewer for pointing this out. We have modified this part as below:

In the revised manuscript, ‘Introduction’ section (clean version Page 5):

‘Epidural analgesia is considered the gold standard of intrapartum analgesia¹, with approximately 10%-83% of women giving birth under neuraxial anesthesia². The rate of cesarean delivery varied from 26% to 58%, exceeding the World Health Organization’s recommendation range of 10% to 15%^{3, 4}.’

Comment 3: A second major problem is that even if recommendations are adequate, care often remains substandard and this is not the fault of recommendations. For example, it is often stated that many years (up to 17 years) elapse between a recommendation is published and its implementation in usual care. Another example can be found in the plethora of studies which compare the content of recommendations and their clinical application. A major gap has been consistently shown, with often only around 50 % of patients receiving adequate care. To paraphrase Lucian Leape, there is more to get (i.e. to improve health in the world) by applying what we already know than by spending money in searching new drugs.

This gap is related to another and different problem which relates to dissemination and uptake of recommendations, i.e. implementation. This has led to create a new discipline, i.e. « implementation science » , which has also its own Journal. This misinterpretation comes back again in the Discussion

section (P24) when the authors show that a recommendation about neurological monitoring has been published in the UK but is not well followed in practice.

Response: We appreciate your valuable feedback. We acknowledge the temporal lag between the recommendations and their implementation, during which the quality of care may not meet the recommended standards, even if the guidelines themselves are sound. We contend that the initial strategy to minimize this gap involves the establishment of uniform and standardized guidelines. This rationale underpins our study's objectives. Furthermore, we have underscored the importance of consistent recommendations and the need to expedite the translation of guidelines into clinical practice within the discussion section of our revised manuscript.

In the revised manuscript, 'Discussion' section (clean version Page 24):

'Current clinical practice often overlooks the importance of neuraxial monitoring; a survey conducted in the UK highlighted that only 56% of units had a formal protocol for postoperative neurological assessment, with a few units deferring evaluation until after the first post-delivery day 39. This discrepancy indicates that despite adequate guideline recommendations, clinical care often fails to meet expected standards. Therefore, when well-supported guidelines are introduced, it is crucial to intensify efforts to enhance the clinical adoption process. Shortening the timeframe for unified practice and enhancing the implementation efficacy of these guidelines can be achieved through the development of clinical practice pathways and the organization of training sessions to disseminate guideline updates. Additionally, greater emphasis should be placed on promoting the implementation of these guidelines. As the adage goes, there is much more to be gained in improving global health by applying existing knowledge than by investing in the development of new drugs.'

Comment 4: Third, it is difficult to understand why the authors limited their search to papers published in English and Chinese. Although China is a vast country, recommendations published in Chinese cannot be applied elsewhere because language is too different. If one wants to include Chinese recommendations, why not include also those in Spanish or in French for example?

Response: We are grateful to the reviewer for highlighting this aspect. In the initial draft's literature review section, we predominantly referenced English and Chinese sources. English serves as the lingua franca for international academic discourse, with the majority of scholarly literature being published and accessed through global academic databases such as Medline and Embase. There is also substantial scholarly output from Chinese researchers. This qualification also takes into account the feasibility and accuracy of the study. Nonetheless, we recognize that this approach may overlook significant contributions in other languages such as French, Spanish, or German, potentially omitting pertinent guidelines. This linguistic limitation is acknowledged within our field and has been noted in prior studies (e.g., *BMJ Open* 2019;9:e026677; *Front Neurol* 2021;12:719849). We have discussed this issue in the 'Limitations' section of our manuscript.

Comment 5: Recommendations were searched from the inception of PubMed to the present days. This is very surprising as recommendations published in the eighties are likely very different from those published in 2023, given the flow of new information gathered in the meantime. Differences would not reflect inconsistencies but simply changes in knowledge.

Response: We apologized for not clearly describing the search result. We conducted a systematically search from inception to 2023, while included the latest versions of all international and national/regional clinical practice guidelines and consensus statements for the obstetric anesthetic management. No guidelines or consensus published in the eighties were included. The finally included guidelines and consensus were the latest versions, with publication dates ranging from 2014 to 2023. We have rewritten this part (clean version Page 10):

'Fourteen guidelines and eight consensus statements, all in their latest versions, were identified, with publication dates ranging from 2014 to 2023.'

Comment 6: Looking at some domains of obstetric anaesthesia, it was found surprising that the list used for their review is not complete. As a French physician, I looked if recommendations originating

from France (published in English in Journals which can be found in PubMed) and surprisingly, I could not find any of them. For example, a Guideline which dealt with inadequate anaesthesia during caesarean section was published one year before (i.e. in the same period of time) the UK guideline (your ref 25) (doi: 10.1016/j.accpm.2021.100934. Epub 2021 Aug 13. Erratum in: Anaesth Crit Care Pain Med. 2021 Oct 17;40(6):100954. PMID: 34400388). It would have been interesting to compare their results and see if inconsistencies can be found.

Other French Guidelines (i.e. for example on preeclampsia) have been recently published in English but were not included.

Response: We really appreciate the reviewer's insights on obstetric anesthesia. We reviewed our search results again and found this guide (doi: 10.1016/j.accpm.2021.100934. Epub 2021 Aug 13. Erratum in: Anaesth Crit Care Pain Med. 2021 Oct 17; 40(6):100954. PMID: 34400388) was not identified, we have added to our study, and the corresponding results have been revised in full manuscript. Please see the amendments in our revised manuscript.

Another guideline (doi: 10.1016/j.gofs.2021.11.003) was already present in our search results but was excluded because only the abstracts were in English and the full text was in French, which was an exclusion criterion.

Comment 7: What the authors of the present review call "inconsistencies" are rather related to domains in which scientific work has not yet provided a clear response. In these cases, it is obvious that human factors can lead to somewhat different views. In decision making processes in which uncertainty is high, what is most important is that reasoning follows a clear and logical process and the final decision might differ between teams which have otherwise a high-quality clinical practice.

Response: We greatly value the reviewer's insightful advice. We concur that the term "inconsistencies" may not be the most appropriate to describe the variations in recommendations across the documents reviewed. Our intention was to collate and summarize differing recommendations on the same topic from the included documents. We have replaced "inconsistencies" with "difference" at the indicated place and the other places.

Comment 8: In some domains, national Guidelines exist but in some others, consistency is obvious as only one is "the" recommendation which is accepted all over the world. This is the case for cardiac arrest during pregnancy where the ERC/ILCOR recommendations are used worldwide. Surprisingly, these recommendations are not mentioned.

One domain which is discussed here has recently appeared, i.e. enhanced recovery after surgery (ERAS). It is believed that all the recommendations in this domain are very consistent, although they have slightly evolved in the last ten years, for example with a more recent inclusion of prehabilitation.

Response: We thank the reviewer for pointing this out. In this study, we focused on the anesthetic management for pregnant, thus, some specialty guidelines which was also involved in pregnant were not included. We have mentioned it in Discussion.

In the revised manuscript, 'Discussion' section (clean version Page 26):

'Secondly, we focused on the anesthetic management for pregnant, some specialty guidelines which was also involved in pregnant (i.e. guidelines for cardiac arrest for maternal⁴⁴) were not included.'

Comment 9: When inconsistencies exist, they may also be related to differences in practice or organisations across countries. For example in some countries, blood transfusion is preceded by type and screen whereas in other (such a France), blood is verified through a different procedure.

Response: We concur with the reviewer's observation that inconsistencies in guidelines may reflect variations in practices or organizational structures across different countries. We have incorporated this consideration into the revised manuscript.

In the revised manuscript, 'Discussion' section (clean version Page 23):

'It is important to acknowledge that discrepancies in recommendations could stem from differing practices or organizational approaches internationally. For instance, some countries initiate blood transfusions following a type and screen with different procedures³⁸.'

Comment 10: It is unclear to me how did the authors decide that a given recommendation is poor in terms of “applicability “ and “editorial independence ». Regarding this last domain, I was again surprised as most recommendations were written by scientific societies which independence is high and also because obstetric anaesthesia is poorly funded by industry, reducing the risk of bias.

Response: We really appreciate the reviewer’s careful consideration for ‘applicability and editorial independence’. The editorial independence domain assessed whether the content of guidelines was influenced by the views of the funding body and whether competing interests of guideline development group members were appropriately recorded and addressed. It is essential for the guidelines to explicitly declare that the final recommendations were not influenced by the funding body’s views or interests. Our objective analysis of the included guidelines ensured that the evaluation was not affected by any risk of bias.

Besides, the distribution of AGREE II domain scores in our study is consistent with patterns noted in previous guideline appraisal studies, where documents typically scored higher in the scope and purpose domain and the clarity of presentation domain, but lower in the applicability domain and the editorial independence domain (e.g., BMJ Open 2019;9:e026677; Cardiology 2004;102:61–6; J Hepatol 2017;67:991–8; Qual Saf Health Care 2010;19).

As discussed in our manuscript (clean Page 22), ‘Low scores in the editorial independence domain often stemmed from inadequate disclosure about the influence of funding bodies and conflicts of interest, with 45.5% of documents included in this study failing to provide such essential information.’

Reviewer-2

Comment 1: P12L53: "fetal" instead of "fatal".

Response: We thank the reviewer for pointing this out. We have checked and corrected those errors. We have also done a thorough checking for all the supplementary materials.

Comment 2: P14L25 and P15L56: "more than"... instead of "less than"...

Response: We apologize for the mistake. We have corrected according to the references.

Comment 3: P23L28 I suggest adding the term "Gaps" to the heading: "Gaps, clinical implications and future research".

Response: Suggested change has been made.

VERSION 2 – REVIEW

REVIEWER	Benhamou, Dan Paris south university hospitals, Anesthesia and Intensive Care Medicine
REVIEW RETURNED	01-May-2024

GENERAL COMMENTS	I have now read this revised version of this manuscript. The text is improved and comments from the reviewers have received an answer with some changes included. It is obvious that the present study is the consequence of an intensive search and analysis. Unfortunately, the main background problems remain. Guidelines may present different recommendations and this is especially true in domains of high uncertainty. This would be similar in all fields of Medicine and obstetric anaesthesia is no exception. Only increased knowledge and future research will be able to minimise the discrepancy between guidelines. The other limitation, well highlighted by the authors is related to the subjective analysis when using the AGREE scale. I would be interested to see if an analysis of the same guidelines performed
--

	by different researchers would find similar results as in the present study. In summary, I do not see what is the added value of this manuscript.
--	---

VERSION 2 – AUTHOR RESPONSE

Reviewer-1

Comment 1: I have now read this revised version of this manuscript. The text is improved and comments from the reviewers have received an answer with some changes included. It is obvious that the present study is the consequence of an intensive search and analysis.

Unfortunately, the main background problems remain. Guidelines may present different recommendations and this is especially true in domains of high uncertainty. This would be similar in all fields of Medicine and obstetric anaesthesia is no exception. Only increased knowledge and future research will be able to minimise the discrepancy between guidelines.

Response: We are grateful to the reviewer for highlighting this aspect. In this review, we aimed to assess the quality of guidelines on obstetric anaesthesia using AGREE method and furnishing comprehensive recommendations for practical application of these guidelines in clinic. We concur with the reviewer's comment that the high uncertainty of guidelines needs future research to minimise the discrepancy between guidelines. We have incorporated this consideration into the revised manuscript. In the revised manuscript, 'Gaps, clinical implications and future research' section (clean version Page 23):

Current guidance and consensus documents uniformly endorse thorough perioperative evaluations and tailored anesthesia methodologies for delivery, including cesarean sections. Notwithstanding these overarching agreements, variances particularly in areas characterized by high uncertainty persist, likely stemming from the variable quality of the documents, gaps in current evidence, or reliance on antiquated studies. This phenomenon is commonplace across all medical disciplines, including obstetric anesthesia. It underscores the complexity of clinical decision-making and the evolving nature of medical knowledge. Thus, continuous learning and future research endeavors are essential in narrowing the gap and ensuring optimal patient care in obstetric anesthesia and beyond.

Comment 2: The other limitation, well highlighted by the authors is related to the subjective analysis when using the AGREE scale. I would be interested to see if an analysis of the same guidelines performed by different researchers would find similar results as in the present study.

In summary, I do not see what is the added value of this manuscript.

Response: We really appreciate the reviewer's careful consideration for the AGREE method. The instruction of AGREE recommended that each guideline should be assessed by at least 2 appraisers, and preferably 4, as this will increase the reliability of the assessment. In this review, four independent reviewers individually assessed each item and the scores were combined to calculate the domain scores. We have mentioned it in Method section.

In methods. 'Appraisal of guidance documents' section (clean version Page 8): 'The included documents were evaluated by four independent reviewers (L.H, N.H, L.J., and D.X.C) using the Appraisal of Guidelines for Research and Evaluation (AGREE) II tool.'